

# Multi-scale analysis of the characteristics of the changing landscape of the typical mountainous region of Southwest China over the past 40 years

Fang Liu[1,2,3,*], Wanbin Wang[2,*], Jinliang Wang[1,3], Xingzi Zhang[2], Jing Ren[1,2] and Yuexiong Liu[2]

[1] Faculty of Geography, Yunnan Normal University, Kunming, Yunnan, China
[2] Yunnan Institute of Environmental Sciences, Kunming, Yunnan, China
[3] Key Laboratory of Resources and Environmental Remote Sensing for Universities in Yunnan, Yunnan Normal University, Kunming, Yunnan, China
[*] These authors contributed equally to this work.

Corresponding author
Jinliang Wang, jlwang@ynnu.edu.cn

## ABSTRACT

**Context**. Yunnan Province is an important ecological security barrier in China. This study investigated the temporal and spatial changes to landscape ecology and is of great significance for guiding landscape protection and future socio-economic development.
**Objectives**. To analyze the temporal and spatial changes of the landscape patterns at the county, river basin, and provincial levels, and clarify and describe the temporal and spatial differentiation characteristics of the landscape patterns in Yunnan Province.
**Methods**. Based on landscape ecology, GIS spatial analysis, and spatio-temporal change analysis, nine landscape pattern indices, and spatial autocorrelation for different years, were calculated.
**Results**. The landscape of Yunnan Province has evolved as a whole toward isolation. The indices of separation and fragmentation changed significantly from 2010 to 2015. From 2015 to 2018 the rate of fragmentation decreased. Fragmentation in the Nu Jiang and Irrawaddy River basins was less than in other basins. The landscape patterns of the Jinsha and Pearl River basins were relatively fragmented due to human activity, socioeconomic development, and utilization. The differences between the Lancang and Red River Basins were relatively small and at an intermediate level.
**Conclusions**. Spatial autocorrelation analysis indicated that there are three areas with typical clusters, namely the Hengduan Mountains where the degree of fragmentation of the landscape was low, while landscape connectivity and aggregation were high. The subtropical region of Southern Yunnan displayed high landscape heterogeneity, a complex shape index, and high connectivity and sprawl. Central Yunnan exhibited a fragmented landscape with poor connectivity and aggregation. These three regions correspond with "the three screens and two belts" in the Main Functional Planning Area of Yunnan Province.

## INTRODUCTION

Landscape patterns are the product of many processes within the landscape over time. They also directly affect the process of landscape creation. A landscape index is a quantitative measure of landscape structural characteristics and change. The theory, method, and application of the indices are at the core of landscape ecology research (*Li et al., 2004*; *Zhang, Fu & Chen, 2003*). In the late 1980s, researchers, such as O'Neill and Turner developed the first landscape indices (*O'Neill et al., 1988*; *Turner & Ruscher, 1988*), while others have constantly improved and developed new landscape indicators. The design of landscape indices incorporates three levels. After decades of development, mature landscape index calculation systems and software now exist. Over the past 10 years, the research on landscape indices has focused on applications. The landscape pattern indices have been used for the analysis of river basins and ecosystem degradation, urban landscape planning, among others (*Pan & Liu, 2016*; *Song et al., 2003*; *Ding & Liang, 2004*; *Chen, Fu & Wang, 2001*; *Zeng et al., 2000*).

Yunnan Province, a region among the richest in biodiversity in the southwest part of China, is considered an ecological security barrier, although few studies have so far been published on the temporal and spatial evolution of its landscape patterns. Several studies have described the Lancang, Nu Jiang, and Red River basins, e.g., *Gan, He & Dang (2003)* while *Jiang, Gao & Ou (2006)*, amongst others, focused on land use and landscape pattern analysis in the Lancang River basin. The results indicate that the Lancang River Basin has rich variety of landscape types and overall uneven distribution of landscape patch density. Cultivated land has the greatest impact on landscape patterns (*Gan, He & Dang, 2003*; *Jiang, Gao & Ou, 2006*). *Ding et al. (2017)* studied the evolution of landscape patterns in the Red River Basin, demonstrating that fragmentation decreased, SHDI and SHEI increased significantly, with landscape tending to be homogeneous. *Zou et al. (2000)* analyzed land use and its effect on the landscape in the Nu Jiang Basin, demonstrating that heterogeneity of the landscape had increased, having previously been homogenous. The Jinsha, Pearl, and Irrawaddy River basins have not yet been studied in terms of landscape patterns. Furthermore, landscape patterns of the six river basins in Yunnan Province have not been compared. Therefore, the present study is of particular importance in this respect.

The current study utilized Land Use Cover Change (LUCC) data in Yunnan Province from 1980 to 2018. The objective was to analyze spatio-temporal variations in landscape patterns in Yunnan Province over nearly 40 years, using landscape pattern indices and spatial analysis within a geographical information system (GIS) environment. Variations in the spatio-temporal characteristics at different scales were investigated (*O'Neill, Hunsaker & Timmins, 1996*; *Saura & Martinez-Millan, 2001*). As it is among the most biodiverse regions in southwest China, and an ecological security barrier, Yunnan Province represents an ideal study area. With the help of GIS spatial autocorrelation analysis, the study identified parameters for variations in landscape patterns in the Province through quantitative analysis of variations in the temporal and spatial characteristics of the landscape patterns, thereby providing an important baseline for subsequent research on landscape and ecological

security, thereby establishing a basis for the development, planning, and management of different river basins in Yunnan Province.

## MATERIALS & METHODS

### Research area

Yunnan Province is located in southwestern China, bordering Myanmar, Laos, and Vietnam, The elevation descends gradually from the northwest to the southeast. The highest point is Kawagebo, the principal peak of Meili Snow Mountain, at an altitude of 6,740 m. The lowest altitude is 76.4 m. Because of the broad effects of geographical conditions and altitude, and the influence of the monsoon circulation, the region has a monsoon climate with small temperature differences in the low latitude mountain region, with distinct dry and wet seasons that change significantly with elevation. There are 7 types of climate in the region: north subtropical, south subtropical, middle subtropical, north temperate, south temperate, middle temperate, and cold temperate. The alpine regions with deep valleys have vertical climatic zones and microclimates. The special geographical location and complex natural environment enhance this extremely rich biological resource. Yunnan Province is one of 17 key areas of biodiversity in China and among the 34 most abundant regions of the world for numbers of species (*Chen & Tang, 2009*). The geographical location and the six river basins in Yunnan Province are presented in Fig. 1.

Yunnan Province has several major international rivers, the Nu Jiang (Salween), Irrawaddy, Lancang (Mekong River), and Red Rivers. It also includes sections of two internal rivers, the Jinsha River and Pearl River. The Jinsha River enters Yunnan from the northwest and forms a V-shape from the northeast to the Yangtze River, then flows into the East China Sea. The Pearl River originates from northeast Yunnan and flows from the northwest to the southeast to Guangzhou and into the South China Sea. Honghe, also known as the Red River, originates in the initial section of Ailao Mountain, passing through Vietnam, then flowing into the South China Sea. Lancang River (Upper Mekong) originates in the Qinghai-Tibet Plateau then flows from north to south, representing the boundary between Laos and Myanmar, after which it flows into the South China Sea in Vietnam (*Wang et al., 2008*). Both the Nu Jiang and Irrawaddy Rivers originate in the Qinghai-Tibet Plateau and flow from north to south into Myanmar, then finally into the Andaman Sea in the Indian Ocean (*Li et al., 2018*).

### Data and sources

Land use data for Yunnan Province from 1980 to 2018 were obtained from the Data Center for Resources and Environmental Sciences, Chinese Academy of Sciences (RESDC; http://www.resdc.cn). The land use/land cover remote sensing monitoring data classification system has adopted a three-level classification system. The first-level category is divided into six subcategories for cultivated land, forest land, grassland, water area, construction land, and unused land, with 25 second-level categories. In the present study, land use data from seven periods were used, including from 1980, 1990, 2000, 2005, 2010, 2015, and 2018. A scale of 1:100,000 was used (*Xu et al.,*

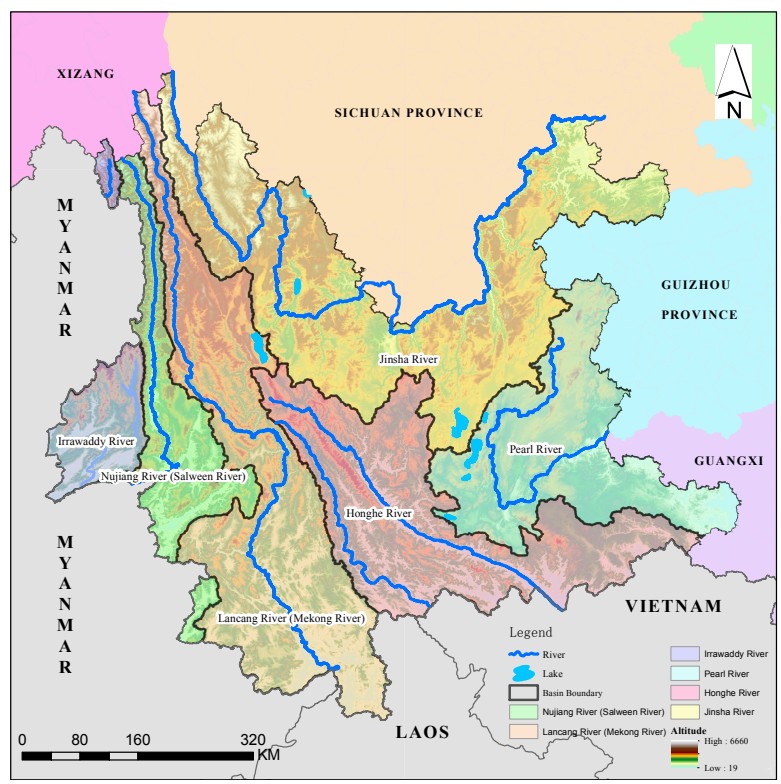

**Figure 1** **Location map of Yunnan Province.** Map credit: https://data.humdata.org/dataset/china-administrative-boundaries, Creative Commons Attribution for Intergovernmental Organisations (https://creativecommons.org/licenses/by/3.0/igo/legalcode).

*2018*). The declared data accuracy for cultivated land and rural residential areas was not less than 95%, that of grassland, woodland, and areas of water was not less than 90%, while the accuracy of unused land was declared to be not less than 85%. The product has previously passed a test of accuracy verification through random sampling at field survey sites and the computed land type at that site (*Liu et al., 2005a*; *Liu et al., 2005b*). The accuracy of the data has satisfied the requirements of macro landscape pattern analysis over a large area (*Zhao et al., 2009*). Vector data from the Centre for Humanitarian Data were used for the administrative region of Yunnan Province, county district (https://data.humdata.org/dataset/china-administrative-boundaries). A digital elevation model was produced at a spatial resolution of 30m from ASTER GDEM V2 data, downloaded from the Geoscientific data cloud platform of the Computer Network Information Center, Chinese Academy of Sciences (http://www.gscloud.cn/sources/).

## Landscape classes

The concepts of landscape, LUCC, landscape classes, landscape patterns, and the relationship between them can be defined as follows. Landscape is a regional spatial entity with particular natural and cultural characteristics, with a clear spatial scope and boundary (*Fu, 2001*). Land cover is either natural or man-made. Land use is a process of

transformation of the natural ecosystem of land into an artificial ecosystem, in which natural, economic, and social factors are combined. Land use/land cover change is abbreviated to LUCC, currently a principal cause of global change and the core focus of research (*Xu et al., 2018*). Analyzing the implication of LUCC from the perspective of landscape ecology is equivalent to the study of a narrow definition of landscape type. Land use structure is equivalent to the type of structures characteristic of landscape patterns and the concrete manifestation of landscape heterogeneity in space (*Gao, 2010*). Landscape pattern refers to the spatial arrangement and combination of landscape elements of different sizes and shapes, including the type, number, spatial distribution, and configuration of landscape constituent units (*Wu, 2000*).

The land resource classification system of the Chinese Academy of Sciences was used to classify landscape types. Level I is divided into six categories, namely cultivated land, woodland, grassland, areas of water, construction land, and unused land, based principally on land resources and their utilization properties. Level II defines the nature of the land resource, and consists of 25 subtypes: cultivated land (nonirrigated farmland, paddy), forest land (shrub, open woodland, other), grassland (high coverage grass, medium coverage grass, low coverage grassland), water (canals, lakes, reservoirs, pits, permanent glaciers and snow, tidal flats), urban multi-industrial and mining residential land (urban land, rural residential area, other construction land), unused land (sandy land, saline-alkali land, marshland, bare land, bare rock stony land, others) (*Xu et al., 2018*; *Peng et al., 2006*). The secondary classification of land use in Yunnan Province in 2000, 2005, 2010, 2015, and 2018 is displayed in Fig. 2.

The classification accuracy and total accuracy were evaluated through a confusion matrix. The data has a comprehensive evaluation accuracy of >94.3% for Level I, and comprehensive accuracy of >91.2% for Level II (*Liu et al., 2018*). Of the land use types in Yunnan Province, forest land is of greatest abundance, while the central part of the province consists of construction land and cultivated land. The distribution has strong horizontal and vertical zonality. There are permanent glaciers and snow in the high-altitude areas of the northwest and scattered bare rocks in the southeast due to rocky desertification. The proportion of the landscape types in 2018 is as follows. Woodland accounts for the greatest proportion at 57.34%, followed by grassland at 22.44%, cultivated land at 17.65%, construction land at 1.17%, areas of water at 0.99%, while unused land only accounted for 0.41%.

## Landscape pattern analysis

In the present study, nine landscape indices in landscape levels were selected and divided into four subcategories of metrics: area/edge, shape, aggregation, and diversity (*Haines-Young & Chopping, 1996*; *Li, Wang & Rong, 2005*). Area and edge metrics represent a loose collection of metrics that describe the size of patches and the length of edge created by these patches. The number of patches (NP) and patch density (PD) were two indices examined. NP reflects the spatial pattern of the landscape and PD is a basic index for landscape pattern analysis (*Wang et al., 2002*; *Qi et al., 2009*). In terms of shape metrics, the interaction of patch shape and size can influence important ecological processes

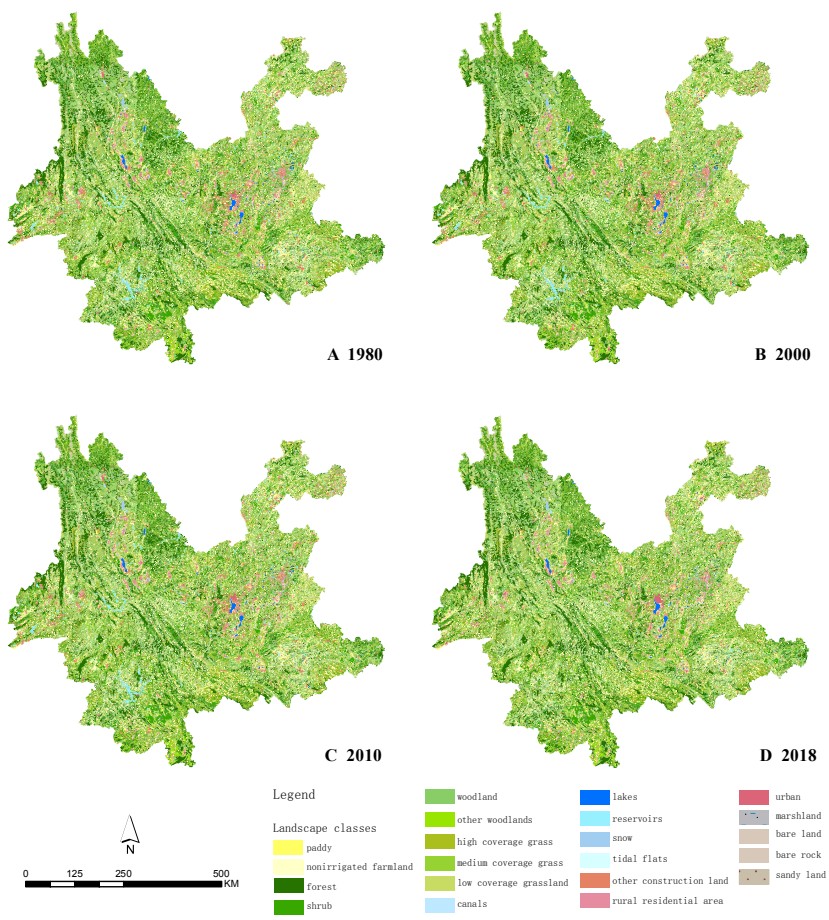

**Figure 2** **Secondary classification of land use in Yunnan Province.** (A) 1980 land use in Yunnan Province. (B) 2000 land use in Yunnan Province. (C) 2010 land use in Yunnan Province. (D) 2018 land use in Yunnan Province. Map credit: https://data.humdata.org/dataset/china-administrative-boundaries, Creative Commons Attribution for Intergovernmental Organisations (https://creativecommons.org/licenses/by/3.0/igo/legalcode).

(*Matthew & Wu, 2002*; *David et al., 2004*). Three indices were selected, namely landscape shape index (LSI), contagion index (CONTAG), and perimeter-area fractal dimension (PAFRAC). The appeal of fractal analysis is that it can be applied to spatial features over a wide variety of scales (*Li et al., 2004*; *Guo et al., 2018*). Aggregation metrics refer to the trend in spatial aggregation of patch types, represented by splitting index (SPLIT) and aggregation index (AI) (*Chen, Xiao & Li, 2002*; *Huang et al., 2012*). Diversity metrics originally gained popularity as measures of diversity in plant and animal species. Measures of diversity are influenced by richness and the variation in values. The two selected indices of diversity metrics were Shannon's diversity index (SHDI) and Shannon's evenness index (SHEI) (*Gong et al., 2008*; *Abdolalizadeh, Ebrahimi & Mostafazadeh, 2019*). Table 1 displays a summary of the selected landscape indices. Additional descriptions can be found in the FRAGSTATS users' guide (*McGarigal, 2015*).

**Table 1 Landscape index calculation formula and ecological significance.**

| Index name | Formula | Index full name | Description |
|---|---|---|---|
| NP | $NP = P$ | Number of patches | NP>0 Describe the heterogeneity of the entire landscape spatial pattern, which is positively related to the fragmentation of the landscape. |
| PD | $PD = \frac{N}{A}(10,000)(100)$ | Patch density | PD>0 Reflecting the integrity and fragmentation of the landscape, a high value indicates low connectivity and a relatively fragmented landscape. Otherwise, the landscape is concentrated and contiguous. |
| LSI | $LSI = \frac{.25E^*}{\sqrt{A}}$ | Landscape shape index | LSI $\geq$ 1 A standardized measure of total edge or edge density to describe the complexity and variation of the shape. |
| SPLIT | $SPLIT = \frac{A^2}{\sum_{i=1}^{m}\sum_{j=1}^{n}a_{ij}^2}$ | Splitting index | $1 \leq$ SPLIT $\leq$ number of cells in the landscape area squared As the landscape is more and more subdivided into smaller patches, the SPLIT value increases, and when the landscape is subdivided to the maximum, it reaches the maximum value when each cell is a separate patch. |
| PAFRAC | $PAFRAC = \frac{\left[N\sum_{i=1}^{m}\sum_{j=1}^{n}\left(\ln p_{ij}\cdot\ln a_{ij}\right)\right]-\left[\left(\sum_{i=1}^{m}\sum_{j=1}^{n}\ln p_{ij}\right)\left(\sum_{i=1}^{m}\sum_{j=1}^{n}\ln a_{ij}\right)\right]}{\left(N\sum_{i=1}^{m}\sum_{j=1}^{n}\ln p_{ij}^2\right)-\left(\sum_{i=1}^{m}\sum_{j=1}^{n}\ln p_{ij}\right)^2}$ | Perimeter-area Fractal dimension | $1 \leq$ PAFRAC $\leq 2$ Describe the complexity of shapes at different spatial scales. For shapes with very simple perimeters, PAFRAC is close to 1, and for shapes with highly tortuous plane-filled perimeters, PAFRAC is close to 2. |
| SHDI | $SHDI = -\sum_{i=1}^{m}\left(P_i^{\bullet}\ln P_i\right)$ | Shannon's diversity index | SHDI $\geq$ 0 Describing landscape diversity and heterogeneity, it is more sensitive to the uneven distribution of various patch types in the landscape. When the landscape contains only 1 patch (no diversity), SHDI $= 0$. |

**Table 1** (*continued*)

| Index name | Formula | Index full name | Description |
|---|---|---|---|
| SHEI | $SHEI = \frac{-\sum_{i=1}^{m}\left(P_i^a \ln P_i\right)}{\ln m}$ | Shannon's evenness index | $0 \leqq SHEI \leqq 1$ Describe the evenness of the distribution of the components of the landscape. Evenness is a supplement to dominance. When the value is small, the dominance is high, and when it approaches 1, the dominance is low. |
| CONTAG | $CONTAG = \left[1 + \frac{\sum_{i=1}^{m}\sum_{k=1}^{m}\left[P_i^\bullet \frac{g_{ik}}{\sum_{k=1}^{m}g_{ik}}\right]\circ\left[\ln\left(P_i^\bullet \frac{g_{ik}}{\sum_{k=1}^{m}g_{ik}}\right)\right]}{2\ln(m)}\right](100)$ | Contagion index | $0 < CONTAG \leqq 100$ Describe the degree of agglomeration or spreading trend of different patch types in the landscape. A high value indicates that a certain dominant patch type in the landscape forms agglomerations and has good connectivity. On the contrary, the degree of fragmentation and fragmentation of the landscape is high. |
| AI | $AI = \left[\sum_{i=1}^{m}\left(\frac{g_{ii}}{\max \to g_{ii}}\right)P_i\right](100)$ | Aggregation index | $0 \leqq AI \leqq 100$ Describe the degree of landscape aggregation, when the landscape gradually converges, the AI value increases. If the landscape is composed of a single patch, AI is equal to 100. When AI is 0, it means that a certain feature type is randomly scattered in the landscape. |

Nine landscape pattern indices of Yunnan Province from 1980, 1990, 2000, 2005, 2010, 2015, and 2018 were calculated using Fragstats 4.2 software at three scales, respectively: province, drainage basin, and county.

## Spatial correlation analysis

Spatial analysis is conducted using a series of techniques for analyzing geographic events, the results of which depend on the spatial distribution of events. A comparison of the spatial statistical characteristics reflects the change in gradients in landscape patterns (*Zeng et al., 2000*). This analysis is based on the distribution of landscape components. The change in landscape heterogeneity and the hierarchical structure of landscape patterns were analyzed. Common spatial analysis methods include spatial autocorrelation analysis, semi-variance analysis, spectral analysis, trend surface analysis, and Kring interpolation.

In the present study, spatial autocorrelation analysis was used to analyze the heterogeneity of the county-scale landscape pattern indices in Yunnan Province. The first law of geography states that everything on the earth is connected. The closer things are, the stronger the connection. Due to the interdependence of spatial data, spatial autocorrelation has become an important content of exploratory spatial data analysis (*Gao, 2010*). Spatial autocorrelation refers to the degree of correlation of a certain attribute of a geographical

parameter in different spatial locations (*Getis & Ord, 2010*). The statistical content includes the spatial relationship and attribute characteristics of the object and its adjacent units. The purpose of spatial autocorrelation analysis is to determine whether a certain variable is spatially correlated and the scale of the correlation. It is often used for a quantitative description of the spatial dependence of objects. A spatial autocorrelation coefficient was used to measure the spatial distribution characteristics of physical or ecological variables and their influence on the neighborhood. Where the values of variables become more similar as the measurement distance decreases, variables are positively spatially correlated. Where the measured value varies more with distance, it is negatively spatially correlated. If the measured values do not show any spatial dependence, then the variable displays spatial irrelevance or spatial randomness.

Local autocorrelation analysis can help determine the heterogeneity characteristics of spatial properties. The degree of correlation between each spatial unit and adjacent units for certain attributes was calculated (*Wu et al., 2015*). The spatial location and scope of the aggregation region were identified, then displayed using Moran scatter and Local indicators of spatial association (LISA) cluster diagrams. The spatial autocorrelation method can visually display the results on a map, clearly showing the differences in the development of the landscape patterns of the counties in the area of study and the relationship between the different types of landscape indices in the local area, and more accurately grasp the heterogeneity of spatial element characteristics. Spatial autocorrelation is an appropriate method for studying spatial phenomena, and it can play an important role in the study of issues related to regional landscape patterns (*Li et al., 2011*). The scatter plots in the quadrants of the Moran scatter diagrams were combined with the level of significance ($P < 0.05$) to obtain a LISA cluster diagram. LISA diagrams can be used as indicators of similarity between the degree of difference and significance for spatial unit attributes and surrounding units. An aggregation diagram represents four different categories of spatial autocorrelation: High-High Cluster, Low-Low Cluster, Low-High Outlier, or High-Low Outlier. According to the LISA diagram, the differentiation of spatial units can be quantitatively analyzed and the distribution pattern of local spatial correlation observed.

## RESULTS AND DISCUSSIONS

### Temporal changes in landscape patterns at a macro level in Yunnan Province

Analysis of temporal changes in 9 landscape indices in Yunnan Province in 1980, 1990, 2000, 2005, 2010, 2015, and 2018 were measured, as shown in Fig. 3. A significant difference was observed in all indices of landscape over different years, and single-factor analysis of variance of the difference between two comparisons was conducted, using a least significant difference (LSD) method, in which the significance of each landscape index was calculated between different years ($P < 0.05$ represented a statistically significant difference).

The annual trend in variation of NP, PD, LSI, and CONTAG were similar. All demonstrated a downward trend from 1980 to 2015, with a slight increase from 2015

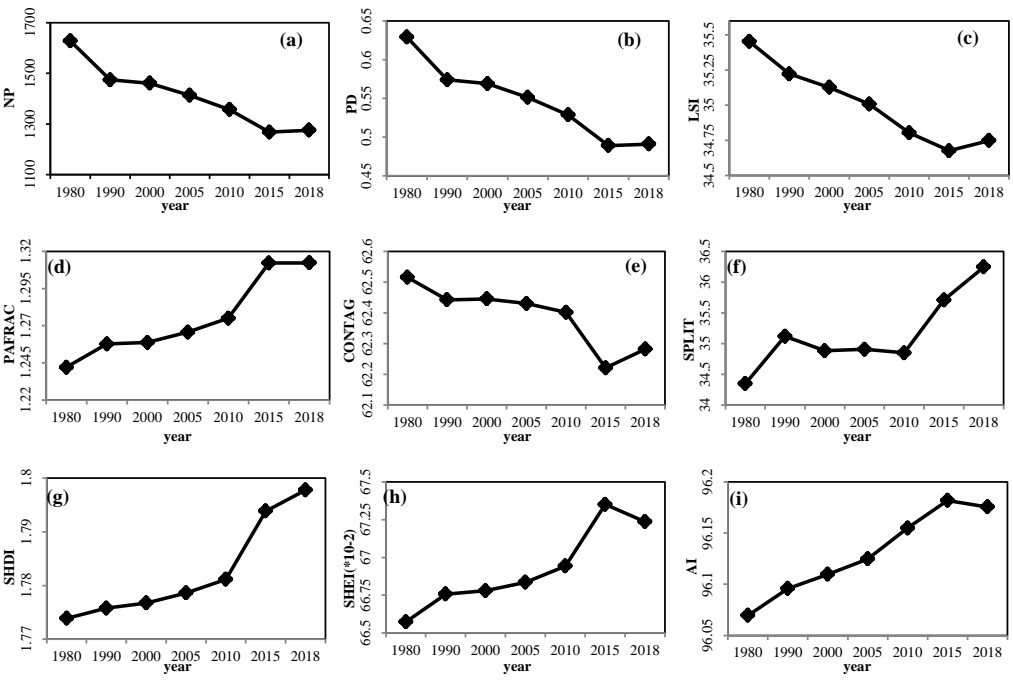

**Figure 3** **Landscape indices in Yunnan from 1980 to 2018.** (A) NP (B) PD. (C) LSI. (D) PAFRAC (E) CONTAG (F) SPLIT (G) SHDI (H) SHEI (I) AI.

to 2018. The interannual changes in LSI and CONTAG indices were not significant ($P > 0.05$). The differences in the PD index between 2005, 2010, 2015, 2018 with 1980 were all significant, as were the differences between 2015, 2018 with 1990, 2000, and 2005, while the differences between the other years were not. From 1980 to 2015, the heterogeneity, fragmentation, and reunion of the landscape in Yunnan Province decreased on the whole, while the complexity of landscape shape decreased. However, from 2015 to 2018, the heterogeneity, fragmentation, and degree of aggregation increased, and landscape shape complexity increased.

The changes in PAFRAC varied from 1980 to 2015. There was a large increase from 2010 to 2015, but it was essentially stable from 2015 to 2018. Differences in both 1980 and annual changes were significant. The difference between 2015 and 2018 was significant compared with 1990, 2000, 2005, and 2010. The differences between other years were not significant. From 1980 to 2018, the overall complexity of landscape patterns in Yunnan Province increased. Significant changes occurred from 2010 to 2015, while the change from 2015 to 2018 was not significant.

The trend in the change in SPLIT from 1980 to 1990 was to increase, with relatively stable values from 1990 to 2010, and sharp increases from 2010 to 2018. The results of significance analysis indicated that the annual change in the SPLIT index was not significant ($P > 0.05$). From 1980 to 2018, the degree of separation of the landscape patterns in Yunnan Province displayed an increasing trend.

There was an upward trend in SHDI from 1980 to 2018. From 1980 to 2010, it rose steadily, and significantly from 2010 to 2015. The results of significance analysis demonstrated that the annual change in the SPLIT index was not significant ($P > 0.05$). From 1980 to 2018, the diversity and heterogeneity of landscape patterns in Yunnan province increased.

The trend in change in SHEI and AI were similar: From 1980 to 2015, they displayed an upward trend, falling slightly from 2015 to 2018. However, the SHEI index rose significantly from 2010 to 2015. The results of significance analysis indicated that the interannual changes in SHEI and AI indices were not significant ($P > 0.05$). In Yunnan Province, the uniformity and cohesion of the landscape increased, while the dominance of the landscape declined.

## Analysis of temporal changes in landscape patterns at the drainage basin level

For the six river basins in Yunnan Province and time change analysis of nine indices in 1980, 1990, 2000, 2005, 2010, 2015, and 2018, homogeneity of variance was not observed with data that was not normally distributed when conducting significance analysis of all the river basin landscape indices. By using nonparametric statistics and a Kruskal-Wallis test, significance between the landscape indices of the river basins was observed. The results of the landscape pattern indices of the six major river basins in the province from 1980 to 2018 are displayed in Fig. 4. Changes in the river basin scale and significance of each landscape index were compared, and the possible reasons for the spatial differences analyzed.

As an index of heterogeneity, PD displayed a slowly decreasing trend in the six river basins, indicating that fragmentation of the landscape decreased and heterogeneity weakened. It is due to the concentration of construction land in the process of urbanization, which reduces the overall fragmentation, although the extent was not large. The most fragmented river basin was the Nu River, followed by the Jinsha River basin, and the least was the Irrawaddy River basin. The value of the Nu Jiang basin was much higher than the others. The high degree of fragmentation in the Nu Jiang basin was due to the topographical influence of its mountains and valleys. The Jinsha River basin had a relatively high degree of fragmentation, largely reflecting socio-economic development.

As indices of diversity and dominance, SHDI and SHEI displayed a slow upward trend, indicating that the landscape abundance of each basin increased, while dominance decreased, and the landscape structure tended to be diverse and homogeneous. This suggests that human economic and social activities in various river basins over the past 40 years have affected the landscape pattern. Of the different basins, the values for the Nu and Irrawaddy River basins were larger, and those of the Lancang River and Red River basins were smaller, indicating that the landscape indices of the Nu Jiang and Irrawaddy River basins were less affected by socio-economic effects.

As indices of connectivity, SPLIT and CONTANG displayed a slow upward trend, indicating that the connectivity between the landscapes of the six River basins had declined because the construction of transportation infrastructures such as highways and railways had caused the landscape to be isolated and the connectivity to decrease. The Jinsha River

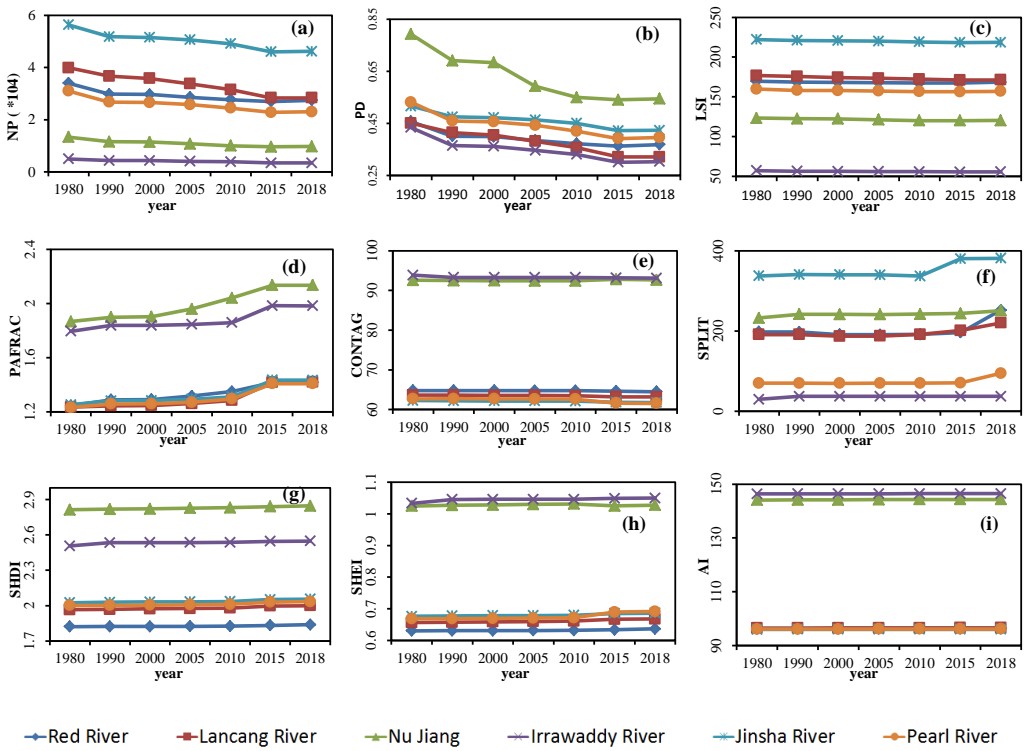

**Figure 4 Change in landscape indices in the Yunnan Basin from 1980 to 2018.** (A) NP (B) PD. (C) LSI. (D) PAFRAC (E) CONTAG (F) SPLIT (G) SHDI (H) SHEI (I) AI.

Basin has the lowest connectivity, and the greatest was the Irrawaddy River Basin, indicating that the intensity of development of transportation infrastructure in the Jinsha River Basin is greater than in other basins.

As indices of convergence and dispersion, AI and PAFRAC first increased and then decreased. The turning point occurred in 2015, but the overall trend increased. This demonstrates that the degree of landscape aggregation increased, but the dominant position of the dominant landscape was less different than other landscapes, with landscape structure tending to be homogeneous. Due to the different levels of development in each basin, and the policy of returning farmland to forest and grassland, the landscape as a whole has evolved from scattered to concentrated. The most concentrated basin was the Irrawaddy River, and the least was the Jinsha River Basin.

The LSI indicator, which characterizes shape, displayed a slow downward trend, indicating that the shape of the landscape in each river basin gradually changed from complex to regular. This was due to the more orderly development and utilization of land by humans and the more regular boundaries of towns and farmland. The six basins with the greatest degree of complex shapes was the Jinsha River basin, and the smallest was the Irrawaddy River basin, indicating that human activity in the Jinsha River basin had caused the greatest disturbance to the landscape.

The landscape indices of the six major basins can be divided into two levels, the first involving the Nu Jiang and Irrawaddy River basins, and the second including the Jinsha, Pearl, Lancang, and Red River basins. The results of significance analysis indicate that the Nu Jiang and Irrawaddy River basins were significantly different for the majority of indices, followed by the Jinsha River basin.

In the first level for landscape diversity, evenness, and connectivity were the Nu Jiang and Irrawaddy River basins, superior to other basins, while the degree of separation, shape indices and other indices were generally lower than other basins. For the Nu Jiang and Irrawaddy River basins, landscape patterns were relatively higher than other river basins, development and degree of utilization being relatively weak, as affected by human activity. However, the valve of PD in the Nu Jiang basin was relatively high, possibly due to the large topographic relief, the topography of the surface leading to landscape fragmentation compared with other basins.

In the second tier, landscape fragmentation, diversity, and heterogeneity of the Jinsha River and the complexity of shape were at a relatively high level, whereas the connectivity and degree of aggregation were relatively low, with the Pearl River basin coming second. For the Jinsha and Pearl River basins, many of the indices were at the same level, indicating that human activity and socioeconomic development in these two river basins had had a great influence on landscape patterns, with development and utilization at a slightly higher level than other rivers. It is worth noting that since 2015, the degree of spread of the Pearl River basin changed from high to low, with evenness indices changing from low to high, compared with the Jinsha River basin, indicating that landscape connectivity and dominance of the Pearl River basin decreased compared with that of the Jinsha River basin from 2015 to 2018. Many indices of the Red River and Lancang River basins were at the same level.

## Spatial and temporal evolution analysis of county-scale landscape indices

Spatial correlation analysis was conducted on 9 landscape indices at the county scale, calculated for 1980, 1990, 2000, 2005, 2010, 2015, and 2018 using the Anselin Local Morans I statistic, from which LISA maps were plotted, as displayed in Fig. 5.

As indices of heterogeneity, the spatial autocorrelation of PD was differentiated and mainly divided into two regions, central and northwestern Yunnan. The High-High Cluster area of Kunming and Qujing city in central Yunnan were highly fragmented and tended to expand outward. The expansion area remained in central Yunnan. The landscape of this area was relatively fragmented and greatly influenced by human society and economy. The Low-Low Cluster area was stable over time and distributed within the Hengduan Mountains in northwest Yunnan. The landscape in this area was less fragmented and the landscape relatively complete and less affected by human activity.

As indices of diversity and homogeneity, SHDI and SHEI displayed no apparent temporal changes, with spatial autocorrelation and differentiation characteristics remaining similar. The High-High Cluster is located in central Yunnan, which has high landscape richness and high homogeneity. The Low-Low Cluster area is located around Ailao mountain which

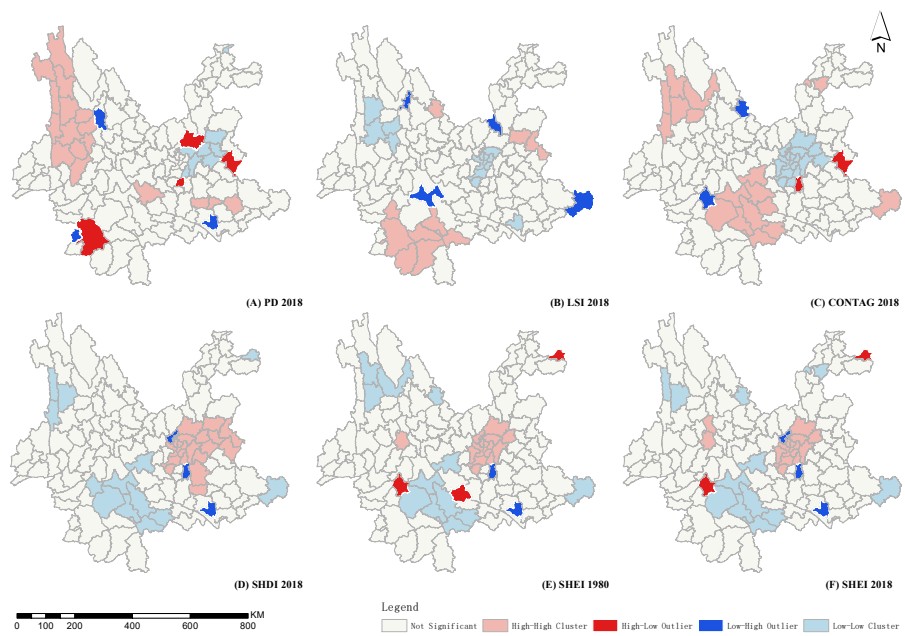

**Figure 5** **LISA diagrams of selected landscape indicators in Yunnan Province.** (A) LISA diagrams of PD 2018. (B) LISA diagrams of LSI 2018. (C) LISA diagrams of CONTAG 2018. (D) LISA diagrams of SHDI 2018. (E) LISA diagrams of SHEI 1980. (F) LISA diagrams of SHEI 2018. Map credit: https://data.humdata.org/dataset/china-administrative-boundaries, Creative Commons Attribution for Intergovernmental Organisations (https://creativecommons.org/licenses/by/3.0/igo/legalcode).

displayed zonal distribution in south Yunnan. This area had a high degree of dominance, with large areas of tropical and subtropical forests. The area was relatively unaffected by human activity, with landscape structure relatively stable.

The CONTAG connectivity index displayed the opposite spatial autocorrelation differentiation characteristics to the diversity and evenness indices, SHDI and SHEI. The High-High Cluster was located in southern Yunnan and Low-Low Cluster area in central Yunnan. The spatial autocorrelation and spatial differentiation of the AI convergence index were similar to those of the CONTAG index in western and central Yunnan, but the inverse in southern Yunnan. This suggests that the poor connectivity and dispersal in central Yunnan result from a greater impact of human activity. Connectivity, aggregation, and dispersion in western Yunnan are greater, and southern Yunnan is better connected but the degree of aggregation did not produce a High-High Cluster area. These two areas have a better natural environment, but southern Yunnan has greater human activity than western Yunnan.

As an index of the shape of the landscape, LSI did not change significantly over time. Spatial autocorrelation characteristics were divided into two areas, western and southern Yunnan. The western Yunnan represented a Low-Low Cluster, with a shape that was not complicated, and a relatively simple landscape system. The southern Yunnan area was a High-High Cluster area with high shape complexity. This area has a subtropical monsoon climate with a more complex landscape system than that of western Yunnan.

The characteristic SPLIT separation index and the fractal dimension index PAFRAC spatial autocorrelation were not apparent, and there were no large-scale contiguous High-High Cluster, Low-Low clustering areas, with High-Low and Low-High singularities appearing only in the marginal counties of Yunnan Province.

The results of spatial autocorrelation analysis of the county landscape indices in Yunnan Province indicate that three regions displayed a change in the apparent characteristics of landscape patterns, namely the longitudinal belt region of the Hengduan Mountains in Western Yunnan, the subtropical area in Southern Yunnan, and the urban area in central Yunnan. These three regions represent the same ecological barriers of the southeastern edge of the Qinghai-Tibet Plateau, Ailao Mountain-Wuliang Mountain, and the southern border of the three barriers and two zones of the "National Major Function Zone Planning" and "Main Function Zoning of Yunnan Province" (*State Council of the People's Republic of China, 2011*). The association has important significance for the construction and protection of the ecological security pattern in Yunnan Province. The characteristics of the landscape patterns of each district are as follows.

The values of PD, LSI, SHEI, SHDI, AI, and CONTAG were clustered in a longitudinal banded region of the Hengduan Mountains in Western Yunnan. The degree of fragmentation of the landscape patterns in this region was relatively low, with shape indices that were not complex, with the degree of landscape connectivity and aggregation that was high. As an ecological barrier region of Yunnan Province, Western Yunnan displayed a natural environment less affected by human activity and the social economy.

The Southern Yunnan subtropical region consists mostly of Pu'er and Xishuangbanna City, with low latitude, abundant water and heat, and high levels of vegetation cover and biodiversity. SHEI and SHDI are in a Low-Low Cluster and NP, LSI, and CONTAG are represented in a High-High Cluster in this region. The degree of fragmentation of the landscape pattern was relatively low, with high landscape heterogeneity, relatively complex shape indices, and a high degree of landscape connectivity and spread. This area is also an important ecological barrier area in Yunnan province, with a good natural environment. However, the influence of human activity and the social economy was greater than that observed in the Hengduan Mountains in Western Yunnan, so heterogeneity was high with relatively complex shape indices.

The urban region in central Yunnan included Kunming, Qujing, and Yuxi City. Values of PD, SHEI, and SHDI represented a High-High Cluster in this region, while AI and CONTAG were Low-Low Clusters. The landscape pattern in this region was broken, with poor landscape connectivity and convergence. This area has a high population density, a developed social economy, with great influence from human activity on the landscape, with fragmentation displaying a trend of gradual expansion. However, some regions still exist with landscape indices around the central Yunnan city cluster that are relatively appropriate. Shape index complexity in this area is relatively low. There are two reasons for this: one is that the urban built-up area results in landscape patches that have a single shape. The other is that the region is a plateau lake aggregation area, with a large water landscape that has only a small range for the change in annual water supply. The overall shape complexity of this area was reduced by these two factors.

In addition, other spatial autocorrelation indices were not significant for this area with a Low-High Outlier with depressions and High-Low Outlier, although not with typical aggregation distribution. The landscape pattern in the central Yunnan province where the landscape was relatively broken up with poor connectivity was Luquan county, while Fumin county remained relatively satisfactory. Furthermore, due to the large area of Lancang County, the spatial pattern was quite different from that of the surrounding areas, with high and low-value depressions observed frequently.

Theoretically, both NP and PD indices reflect the degree of fragmentation of the landscape in the sense of landscape ecology. Our analysis, however, indicates that NP and PD display different spatial correlations. As this study is based on landscape index calculations within the administrative region of Yunnan Province, the area of each administrative region is different, with the NP indices greatly affected by the administrative region, and therefore not reflecting regional landscape heterogeneity and fragmentation characteristics in a particularly accurate way. Spatial correlation analysis demonstrated that PD was more consistent with the actual geography of the region.

## DISCUSSION

To compare the actual situation in Yunnan Province, nine landscape indices were selected in the present study to assess the characteristics of the landscape patterns and evaluate whether they displayed applicability at the province, river basins, and county scales. Of the indices, PD best-reflected differences in the regional natural environment in Yunnan province, with LSI and SPLIT shape indices better reflecting the impact of human activity and the social economy on the landscape. PD better reflected landscape fragmentation than NP.

The landscape patterns of Yunnan Province have been stable although affected by human activity over the past 40 years, with rapid development from 2010 to 2015 leading to significant changes in landscape indices. The Hengduan Mountains in northwestern Yunnan have a low degree of fragmentation, high connectivity, and uncomplicated landscape shapes. This area belongs to the end of the Hengduan Mountains, with low population density and little human activity to disturb the landscape. The urban area in central Yunnan has a high degree of fragmentation, poor aggregation and connectivity, and low dominance. This area is a concentrated urban area with high population density and intense social and economic activity, the factors leading to fragmented landscapes with poor connectivity. The subtropical area of southern Yunnan has high landscape richness, dominance, and spread. The forest system representing the predominant landscape in this area has a commanding effect on the regional landscape pattern. Population density and the level of socio-economic development here are at a moderate level compared with that of western and central Yunnan, with an impact on the landscape that is limited, although not negligible.

We used the same resolution to study different scales, without considering the effects of grain size on landscape indices, which we hope to study in a future exercise, selecting the best resolution for different scales. We only analyzed the spatial–temporal evolution of the

landscape patterns without studying its driving factors. The next step is to analyze whether the spatial–temporal evolution and differentiation of the landscape pattern are affected by socioeconomic factors or natural environmental factors.

## CONCLUSIONS

(1) Over the past 40 years, the diversity and aggregation of the landscape in Yunnan Province have gradually increased, the degree of dominance, connectivity, and shape complexity having declined, while landscape structure has stabilized. Significant changes have occurred in indices such as diversity, dominance, and contagion from 2010 to 2015, the homogenization of the landscape having accelerated, while some indices have declined from 2015 to 2018.

(2) The majority of the landscape indices in the Nu Jiang and Irrawaddy River basins are significantly different from other basins, indicating that these two basins have retained a better natural environment with a stable landscape structure. Compared with other basins, the degree of development and utilization is relatively weak, and they have been less affected by human activity. The landscape indices for the Jinsha and Pearl River basins are at similar levels. Human activity and social and economic development have greatly influenced their landscape patterns, the degree of development and utilization higher than that of other basins. The landscape indices of the Red River and Lancang River basins are at a moderate level, the remaining natural environment and socio-economic development from human activity moderate for Yunnan Province.

(3) Through landscape pattern spatial autocorrelation distribution characteristics, three regions with clearly changed landscape pattern characteristics in Yunnan Province are the longitudinal belt of the Hengduan Mountains in Western Yunnan, the urban area in central Yunnan, and the subtropical area of southern Yunnan. These three regions correspond with ''the three screens and two belts'' in The Main Functional Planning Area of Yunnan Province.

### Funding

This work was supported by Multi-government International Science and Technology Innovation Cooperation Key Project of National Key Research and Development Program of China for the ''Environmental monitoring and assessment of land use/land cover change impact on ecological security using geospatial technologies'' [grant number: 2018YFE0184300], the Key Research and development plans in the field of social development in Yunnan Province for the ''Research and Application Demonstration of Ecological Protection and Restoration Technology in Yunnan Typical Ecological Region'' [grant number 2019BC001], and the Key Program of Basic Research of Yunnan Province of China for the ''Scientific Investigation on the Geographical Environment of the Tourism Health Industry in the Tropic of Cancer (Yunnan Section'' [grant number 2019FA017]. The funders had no role in study design, data collection and analysis, decision to publish, or preparation of the manuscript.

## Grant Disclosures

The following grant information was disclosed by the authors:

National Key Research and Development Program of China: 2018YFE0184300.

Research and Application Demonstration of Ecological Protection and Restoration Technology in Yunnan Typical Ecological Region: 2019BC001.

Scientific Investigation on the Geographical Environment of the Tourism Health Industry in the Tropic of Cancer (Yunnan Section): 2019FA017.

## Competing Interests

The authors declare there are no competing interests.

## Author Contributions

- Fang Liu and Wanbin Wang conceived and designed the experiments, performed the experiments, prepared figures and/or tables, and approved the final draft.
- Jinliang Wang and Xingzi Zhang conceived and designed the experiments, authored or reviewed drafts of the paper, and approved the final draft.
- Jing Ren analyzed the data, authored or reviewed drafts of the paper, and approved the final draft.
- Yuexiong Liu performed the experiments, analyzed the data, prepared figures and/or tables, and approved the final draft.

## Data Availability

Raw measurements are available in Supplemental Files.

## Supplemental Information

Supplemental information for this article can be found online at http://dx.doi.org/10.7717/peerj.10923#supplemental-information.

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
