# Peer review of "Multi-scale analysis of the characteristics of the changing landscape of the typical mountainous region of Southwest China over the past 40 years"

_PeerJ, doi:10.7717/peerj.10923_

## Round 0.1 · original submission · Major Revisions

The reviewers' comments on your work have now been received. The referees acknowledge the potential interest of your work, but between them, they also raise a number of concerns, which must prevent us from offering to publish the paper in its present form. The referees’ reports seem to be quite clear. Naturally, we will need you to address all of the points raised. The writing is not clear enough, and there are some grammatical and structural mistakes in the manuscript.

Reviewer 1 ·

Basic reporting

This paper deals with an important research question in a large mountainous area with unique ecological and geographical characteristics in southwest China, using data from almost 40 years (1980 – 2018). After reviewing the paper, I think it can be published in this journal after some minor/intermediate revisions. I have provided detailed comments in the annotated PDF document.

Experimental design

The experimental design is suitable for the research objective.

Validity of the findings

Multi-level spatial and temporal variations of landscape patterns are revealed in the study area. Reasons or possible reasons behind the variations are also explained. The findings are important contributions to relevant literature in the study area, and can be used as baseline datasets for future studies.

Additional comments

The paper involves a large amount of datasets covering Yunnan Province in southwest China, employs solid methods, and presents significant results with high-quality figures. The paper is publishable after minor/intermediate revisions, mainly in the writing style and some grammatical errors. I have suggested changes in the annotated PDF document, and I’d like to suggest that the authors check the following points throughout the paper:
(1) Line 36, the reference for Turner is missing.
(2) Change the old name “Nu Chiang” to “Nu Jiang” or “Nu River”, and use a consistent name for the river in the paper.
(3) Pay attention to uppercase and lowercase in river names. For example, the following words are used interchangeably in the paper: “Nu Chiang”, “Nu River”, and “Nu river”. Consistent names should be used.
(4) Check if “river basin” should be used in some paragraphs, instead of “river”. For example, when discussing fragmentation in Lines 193-200, “river basin” should be used, not “river”. A river as a linear feature does not have fragmentation, while river basins can be fragmented.
(5) The landscape indices are listed in Table 1. It is suggested to put the full names of the indices in the third column of the table to improve the readability of the paper. For example, for the “NP” index, the full name “Number of patches” should be added to the third column. The title of the third column in Table 1 can be “Description”.

Annotated reviews are not available for download in order to protect the identity of reviewers who chose to remain anonymous.

Reviewer 2 ·

Basic reporting

no comment

Experimental design

no comment

Validity of the findings

no comment

Additional comments

Thank you for the opportunity to review the manuscript “Multi-scale analysis of the characteristics of the changing landscape of the typical mountainous region of southwest China over the past 40 years”. Generally, the academic contribution and innovation of this study are not clear to me. My comments and questions are as follows:

1.Please clearly explain the innovation and contribution of this study.
2.The draft is very long, some parts are not necessary, e.g. Line 103-112, it’s not necessary to describe the secondary categories and this can be included in the 2.2 Data and sources. There are many cases that the long draft is not closely related to the study. Fig.2 didn’t show the secondary classes.
3.The experiments are easy and the data is products. What’s the original data that you produced?
4.Line 97-101 is not precise. The used dataset need to give out direct websites. Projection is not needed.
5.LUCC data is the most base of the analysis of this study, what’s the accuracy of this dataset in Yunnan Province? Please adding some materials about the data accuracy and methods. There are many LUCC datasets, why use this one?
6. Adding a map to show the six river basins.
7.Why used nine indicators in this study? What are their differences, advantages, or disadvantages? How to understand the results, other than just showing the numbers?
8.Line 185-365: The results of the nine indexes are very long. Can the authors summarize the results and give out some main points? Rather than introduce nine indicators one by one.
9.The conclusion needs to be improved.
10.The LUCC in Fig.2 is strange (construction land is too small). Adding the year to the figures and the legends are not clear. The four sub-figures can be put together as one figure.
11.The titles of the figures should be above the figures. The titles can be improved.
12.Table 1 needs citation.

Reviewer 3 ·

Basic reporting

The aim of this manuscript is to analyze temporal and spatial changes of landscape patterns at county, watershed, and provincial levels over past 40 years. It is interesting to address the landscape changing at multi-scale in Yunnan Province, China, one of the most biodiversity richness regions in the world. However, this work needs substantial improvement in several related critical matters. Moreover, the sections of introduction, methodology, results, are needed to be clarity and language editing.

Experimental design

1) The title, “the characteristics of the changing landscape”, what are the characteristics? And what is specific landscape? Forest? Agricultural? Urban? River? Mountain? and so on. Actually, this study is more focus on the land cover changes of Yunnan Province, China at different scales. Thus, the title can be change to “Spatial and temporal changes of land-cover at multi-scale in mountainous region of Southwest China over the past 40 years”

2) In section 1 introduction and literature review, in this section, it is lack of literature review of LUCC across multi-scale. Moreover, the research questions should be well defined. Why the land cover changes should be analyzed at county, watershed, and provincial levels? What is the research rationale? Why 40 years? Again, I think that landscape pattern changes more focus on specific landscape not general landscape.

3) line 43-45, “an important center of biodiversity within the Southwest part of China, represent ecological security barriers”. What is “center of biodiversity”, and why it represent ecological security barriers? The ecological security barriers for what? “The pattern of the landscape within Yunnan Province has not been improved” What is “the landscape has not been improved”? what’s this mean? “

4) line46-51, literature review listed previous studies have addressed the LUCC of the Lancang watershed, Nu Jiang watershed, Red River watershed. However, what are the main findings of these studies? Why the Jinsha, Pearl, and Irrawaddy River basins have not been studies and why do you want to study LUCC for those watershed?

5) line 53-55, these two sentences should be re-written. LUCC is the first time be used in this paper, the full name should be listed.

6) Line 53-65, in this paragraph, authors always mentioned the GIS spatial analysis was used in this study. What specific spatial analysis was used in this study?

7) Line 53-65, in this paragraph, authors always mentioned the GIS spatial analysis was used in this study. What specific spatial analysis was used in this study?

8) Line 92-101, in this paragraph, “the LUCC data for Yunnan Province from 1980 to 2018”, it should be land cover or land use maps, is not LUCC (land cover and land use change). Moreover, the methodology of how to make those land cover or land use maps and the quality (accuracy assessment) of those maps should be provided.

9) Line 102, landscape types, as mentioned above, landscape, land cover, land use, landscape change, LUCC, those concepts are confused. Authors should clarify those concepts, particularly what is landscape types.

10) Line 124, spatial correlation analysis, why? For what?.

Validity of the findings

no comments

Additional comments

Summary, It is interesting to address the landscape changing at multi-scale in Yunnan Province, China, one of the most biodiversity richness regions in the world. However, this work needs be improved, and the language need to be edited by native speaking people.

Annotated reviews are not available for download in order to protect the identity of reviewers who chose to remain anonymous.

---

## Round 0.2 · Minor Revisions

Both referees suggest that the submission may be publishable, but only after some revisions have been made to your manuscript. Therefore, I invite you to respond to their comments (especially those comments from reviewer 2 who reported that their previous comments had not been addressed) and revise your manuscript accordingly.

Reviewer 1 ·

Basic reporting

No comments.

Experimental design

No comments.

Validity of the findings

No comments.

Additional comments

Line 235, remove "Analysis of".
Line 251, use "The changing trend in PAFRAC varies ...." or "The changes in PAFRAC vary ..."
Line 259, insert a space before "stable".
Line 276, replace "bansins" with "basins".
Line 279, replace "biasin" with "basin".

Reviewer 2 ·

Basic reporting

See below

Experimental design

See below

Validity of the findings

See below

Additional comments

Thank you for the opportunity to review the manuscript “Multi-scale analysis of the characteristics of the changing landscape of the typical mountainous region of southwest China over the past 40 years”. Generally, the manuscript has been improved after the reversion, especially the language and expression. My comments and questions are as follows:

1. In the Rebuttal Letter, the authors wrote “There are many LUCC products in China, such as GLOBE 30 by Gong Peng from Tsinghua University”, which is not true. GLOBE 30 is developed by Dr. Jun Chen from National Geomatic Center for China. The reasons such as “It is the Institute of Geography of the Chinese Academy of Sciences. The technical strength of UCAS is at top level in China.” is not very convincing. The authors could have a deeper understanding of the LUCC products themselves.

2. Line 58-66 is verbose. The actual contents of this research are not complicated and it’s fine to emphasis the significance. But the similar expression has been repeated many times, including the manuscript and Rebuttal Letter.

3. Section 2.3 is not necessary, because Section 2.2 has some description. Or the authors can include some contents to Section 2.2.

4. The English needs improved and the grammar of line 140 is not right. Line 144 “more descriptions could be also found in the user’s guide of FRAGSTATS”. Did the authors pick the nine indexes from the FRAGSTATS?

5. It’s a big problem that there is no actual discussion in the Section “Results and discussion”. Generally, it’s only about the results of the nine indexes. How to understand the numbers? What about the comparison between this study and other studies? What’s the policy implication? What are the limitations and future perspective? A lot can be discussion.

6. The conclusion is too long and some are discussion, not inclusion.

7. In Fig.2 the LUCC changes of the four figures are not obvious. Please add some contents about the accuracy of this data in Yunnan Province and the detailed areas of the main LUCC types. Also, there are some strange lines in the figure (e.g. triangle in the snow and lines in the high coverage grass)

---

## Round 0.3 · Minor Revisions

The authors have apparently addressed the reviewers' concerns. The scientific portion, however, has passed review. I would suggest the authors use the PeerJ copyediting service or another similar service. The issues are not such that the work is unreadable (of course), but the paper does not meet normal PeerJ standards and would detract from the message of the paper if published in this state.

---

## Round 0.4 · accepted · Accept

Considering that all issues raised by the reviewers were properly and carefully addressed by the authors with the improvements as required, it is our opinion that the manuscript can be accepted in its present version. Currently, this paper has met normal PeerJ standards.